# Automatic Detection of Near-Surface Targets for Unmanned Aerial Vehicle (UAV) Magnetic Survey

**Yaxin Mu [1,2,3], Xiaojuan Zhang [1,2,*], Wupeng Xie [1,2,3] and Yaoxin Zheng [1,2,3]**

1   Aerospace Information Research Institute, Chinese Academy of Sciences, Beijing 100094, China;
    muyaxin15@mails.ucas.ac.cn (Y.M.); xiewupeng15@mails.ucas.ac.cn (W.X.);
    zhengyaoxin17@mails.ucas.ac.cn (Y.Z.)
2   Key Laboratory of Electromagnetic Radiation and Sensing Technology, Chinese Academy of Sciences,
    Beijing 100190, China
3   School of Electronic, Electrical and Communication Engineering, University of Chinese Academy of Sciences,
    Beijing 100049, China
*   Correspondence: xjzhang@mail.ie.ac.cn; Tel.: +86-10-5888-7276

**Abstract:** Great progress has been made in the integration of Unmanned Aerial Vehicle (UAV) magnetic measurement systems, but the interpretation of UAV magnetic data is facing serious challenges. This paper presents a complete workflow for the detection of the subsurface objects, like Unexploded Ordnance (UXO), by the UAV-borne magnetic survey. The elimination of interference field generated by the drone and an improved Euler deconvolution are emphasized. The quality of UAV magnetic data is limited by the UAV interference field. A compensation method based on the signal correlation is proposed to remove the UAV interference field, which lays the foundation for the subsequent interpretation of UAV magnetic data. An improved Euler deconvolution is developed to estimate the location of underground targets automatically, which is the combination of YOLOv3 (You Only Look Once version 3) and Euler deconvolution. YOLOv3 is a deep convolutional neural network (DCNN)-based image and video detector and it is applied in the context of magnetic survey for the first time, replacing the traditional sliding window. The improved algorithm is more satisfactory for the large-scale UAV-borne magnetic survey because of the simpler and faster workflow, compared with the traditional sliding window (SW)-based Euler method. The field test is conducted and the experimental results show that all procedures in the designed routine is reasonable and effective. The UAV interference field is suppressed significantly with root mean square error 0.5391 nT and the improved Euler deconvolution outperforms the SW Euler deconvolution in terms of positioning accuracy and reducing false targets.

**Keywords:** near-surface targets detection; UAV-borne magnetic survey; UAV interference field; interpretation method of magnetic field; Euler deconvolution; YOLOv3

## 1. Introduction

Magnetic detection is an ancient remote sensing technology that can be traced back to the mid-19th century for the exploration of iron ore deposits. Today, magnetic sensing is still attractive in various applications by virtue of its passive, rapid, noninvasive detection performance [1,2], like geological exploration [3,4], Unexploded Ordnance (UXO) detection [5,6], archaeology [7,8], and the magnetoencephalogram (MEG) and magnetocardiogram (MCG) [9,10] in the field of biomedical science, etc. The traditional handheld magnetic survey can acquire high-density magnetic data, but it is time-consuming, labor-intensive, and limited by the terrain. The fixed-wing airplane-based magnetic survey can realize fast and wide-range measurement, but the spatial resolution of magnetic is low and

only suitable for the detection of large targets. With the advance in the small Unmanned Aerial Vehicle (sUAV), UAV system provides an excellent platform in achieving low-cost, low-altitude, long-range, fast magnetic field monitoring [11–14], compared with the terrestrial, manned aircraft, and satellite magnetic survey, which opens a brand-new era in using UAV-borne magnetic survey.

This paper focuses on the remote detection of near-surface targets, like UXO and steel pipe, using the UAV magnetic measurement system. The UAV magnetic survey shows significant advantages in the magnetic spatial resolution, work efficiency and experimental safety. However, UAV-borne magnetic survey is facing a lot of tough challenges due to its limited load capacity, short battery life, interference field generated by the drone, uncertainty of the position of the magnetic sensors, and so on.

Recently, lots of researchers have devoted themselves to the development of the UAV-magnetometer sensing devices, a variety of miniaturized, lightweight magnetic sensors and acquisition systems have been installed to the drones, different field experiments have also been carried out. Douglas et al. [15] developed an autonomous aeromagnetic system using a rotary-wing UAV with a fluxgate magnetometer to detect the small ferrous minerals. A path planning algorithm based on the lawnmower coverage pattern was proposed to cover the test region efficiently, the magnetic contour map was obtained to reflect the distribution of subsurface anomalous targets. Alireza Malehmir et al. [16] used the DJI S1000 rotary-wing UAV equipped with the GEM 19GW magnetometer, a GPS antenna, and data recorder module to detect the iron-oxide deposits in central Sweden. The field experiment showed that the UAV magnetic measurement system was able to clearly delineate the mineralization with finer sampling, higher resolution over the conventional aeromagnetic survey. Christoph Eck and Benedikt Imbachy [17] designed an aerial magnetic sensing scheme with the Scout B1-100 UAV helicopter and a three-axis magnetic sensor, which was applied in the humanitarian rescue. In [18], two Geometrics G-823A cesium vapor magnetometers, a computer, power supplies, differential GPS, IMU, and fluxgate magnetometer are mounted on the fixed-wing UAV, called Venturer, for the aeromagnetic surveying.

To date, considerable progress has been made in the integration and testing of the UAV-magnetometer system. Whether it is based on the fixed-wing UAV [17,18] or multi-rotor UAV [15,16,19], a single magnetometer [15–17] or multiple magnetometers [18], the UAV-magnetometer system shows excellent performance over the handheld and aeromagnetic devices. However, the research on the processing and interpretation of UAV magnetic data is quite rare and challenging because of the low quality of magnetic data and the lack of interpretation methods for the UAV-based magnetic survey. Most studies attempt to show the distribution of underground anomalies roughly through the 2D magnetic contour map from the UAV magnetic data, but the 3D location of targets cannot be obtained by the available interpretation ways. If a sophisticated processing routine for the UAV magnetic survey is developed, the UAV-borne magnetic survey is bound to gain more attention in scientific research and engineering applications.

One of the biggest obstacles in collecting high-quality magnetic data by the UAV-magnetometer system is the interference magnetic field from the drone [18,19]. The drone system is not a non-magnetic platform, some modules and structures are made of ferromagnetic materials, which have a serious impact on the magnetometers attached to the drone and will degrade the measurement accuracy of system. In fact, compensating the interference field originated from the flight platform is not a new problem. Aeromagnetic compensation has been studied deeply for the traditional manned aerial magnetic survey. The classic magnetic compensation scheme is the Tolles–Lawson model [20,21], but this compensation scheme cannot be used in the drone systems. Because the aeromagnetic compensation flight is performed in a few kilometers of altitude typically, providing a uniform background field. That is not acceptable for the drones due to its general maximum flying height is tens of meters, where the background magnetic field is spatially non-uniform for the existence of geological and cultural sources. Now, a long rope or rod (3–5 m) is suspended below or in front of the drone platform in [15–19] to reduce the effects of the drone on the sensors. However, this

long rod structure is prone to an unbalanced system and low flight stability, which is not an ideal solution. The compensation of UAV interference field is an open problem [18,19] and still requires further research.

Although there have been numerous studies on the processing and interpretation of magnetic field data for the ground-based magnetic survey, these techniques cannot be directly applied to the UAV-based magnetic survey. Marc Munschy et al. [5] put forward a complete processing framework that is based on the handheld multi-sensor fluxgate magnetometers for the UXO detection. The calibration operation was firstly conducted to eliminate nine errors from the fluxgate sensor. Then, analytical signal method and magnetic dipole inversion were adopted to interpret the magnetic data. The analytical signal method is able to estimate the location of objects by the first-order and second order magnetic gradient field, which requires the high signal-to-noise (SNR) magnetic field data [5,22]. The magnetic dipole inversion is capable of determining the position and the 3D magnetic moment vector of the targets at the same time, but the magnetic dipole inversion is not an automatic processing technique for the survey data. It is necessary to manually extract the anomaly field generated by an isolated target, and the nonlinear inversion method depends on the initial value, so the calculation time is long [5,23]. By contrast, Euler deconvolution is more appropriate to achieve automatic interpretation of survey data [24]. A fully automatic detection of UXO for the terrestrial magnetic survey has been proposed by Kristofer Davis et al. [25,26], which was essentially the sliding window (SW)-based Euler deconvolution process. Euler deconvolution is a rapid way to estimate the location of the isolated target, but due to the size of sliding window is changing in an estimated range [25–27], the entire process is complex and cumbersome, increasing the processing time. Furthermore, the result of SW-based Euler method depends on the attenuation of the magnetic field, has nothing to do with the strength of the magnetic field, resulting in a large number of false anomalies, leading to the rise in the number of false targets. Usman et al. [28] proposed a few filter techniques to suppress the unreliable Euler solutions by constraining the amplitude of analytical signal after the SW-based Euler deconvolution.

In this paper, we introduce the basic configuration of UAV-magnetometer system and put forward a complete data processing routine for the UAV-borne magnetic survey, including the elimination of regional background field, the removal of interference field originated from the UAV platform, magnetic mapping, automatic detection and positioning of subsurface targets. Among them, the elimination of UAV interference field and automatic detection and positioning method are highlighted for the UAV-based magnetic survey. Unlike the traditional aeromagnetic compensation scheme based on the attitude information, a two-channel linear time-invariant (LTI) system is modelled [29,30] and the spatial correlation of magnetic field signals sampled by two closely spaced sensors is employed to separate the magnetic anomaly signal and the UAV interference field from the total field signal. Besides, an improved Euler deconvolution method is presented to detect the underground targets automatically, which is the combination of YOLOv3 (You Only Look Once version 3) and Euler deconvolution algorithm. YOLOv3 is a deep convolutional neural network (DCNN) based artificial intelligence algorithm and excels in image and video detection [31]. Here, YOLOv3 is applied in the context of magnetic survey for the first time, replacing the traditional sliding window process, showing better performance in reducing false targets and improving the positioning accuracy. In addition, the improved Euler deconvolution is simpler and faster, which is quite suitable for the large-scale UAV magnetic survey.

The paper is organized as follows. Section 2 presents the configuration of UAV-magnetometer system and the workflow for the UAV-borne magnetic survey, all procedures are briefly explained. Section 3 introduces the key techniques of the magnetic data processing; the compensation of UAV interference field and the improved Euler deconvolution method are described in detail. The field tests and experimental results are discussed in Section 4. Section 5 gives the discussion and remarks regarding the proposed framework. Section 6 provides the conclusions of this research.

## 2. System and Workflow

### 2.1. UAV-Magnetometer System

A novel UAV magnetic measurement system is developed for the detection of near-surface targets. The UAV-Magnetometer system consists of two magnetometers, radar altimeter, differential GPS, data recording module and power module, as shown in Figure 1. The flight platform is the eight-rotor DJI MG1 unmanned aerial vehicle. Two Cs optically pumped magnetometers are mounted on the center of drone by a vertical boom, the vertical distance between two sensors is 0.3 m. The Cs optically pumped magnetometer is a lightweight, small-sized CAS-L3 designed for the drones by the Aerospace Information Research Institute, Chinese Academy of Sciences, China. The operating range of CAS-L3 is 15,000 nT to 105,000 nT and the noise sensitivity is 0.6 pTrms√Hz@1 Hz in the shielded room. The radar altimeter is RD2412R, manufactured by SZ DJI Technology Co., Ltd., China. It measures the height of the drone above the ground ensuring that the UAV can actively avoid obstacles and fly safely in the complex terrain. Differential GPS T300, produced by Shanghai Sinan Satellite Navigation Technology Co., Ltd., is used to acquire the location of UAV, positioning accuracy is on the order of cm. Magnetic field data and position information are synchronized by the pps (pulse per second) signal. The custom-designed data recording module is constructed and fixed on the bottom of the UAV system. All modules have been installed reasonably to ensure the stability and balance of the UAV-magnetometer system; the entire payload weighs 4.37 kg meeting the requirement of the maximum load (13.7 kg).

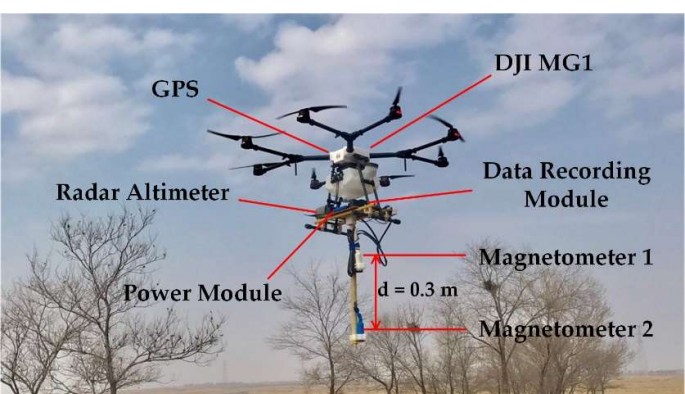

**Figure 1.** The Unmanned Aerial Vehicle (UAV)-magnetometer system, carrying two magnetometers, radar altimeter, GPS, data recording and power module. The top magnetic sensor is labelled as magnetometer 1 and the bottom magnetic sensor is labelled as magnetometer 2.

### 2.2. Workflow for UAV-Borne Magnetic Survey

Applied to the near-surface targets detection, the workflow of the UAV magnetic survey is divided into three stages, the first one is data collection, including the preparation of test area, the setting of flight path and drone; the second phase is data processing, involved some signal processing methods to obtain high quality magnetic field data, the final stage is data interpretation, achieving the automatic detection and position estimation of buried targets. The complete workflow is listed in Figure 2.

The mission planning is the preliminary task of the magnetic survey, comprising the setting of test area, profiles, and the drone. For the planning of the test site, the region of interest, topography, vegetation, weather and other factors need to be evaluated carefully. For the planning of the profile, the flight direction, length, and spacing should be considered. In addition, the working mode of UAV should be selected, such as manual or automatic flight, flying speed, above ground level (AGL), battery life. In short, the perfect mission planning lays the foundation for the UAV magnetic survey, ensuring that the drone can fly normally and collect the magnetic data.

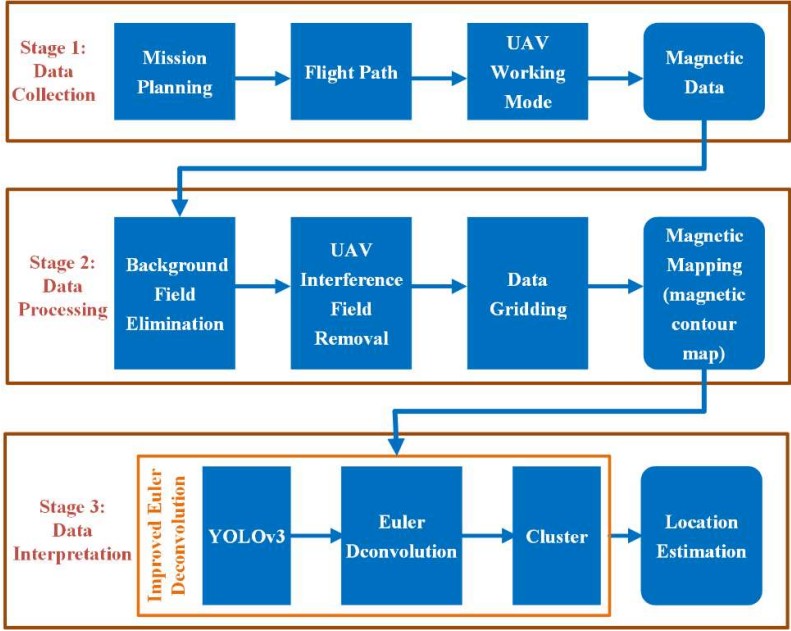

**Figure 2.** The workflow for UAV magnetic survey.

The purpose of data processing is to reduce the noise signal, extract magnetic anomaly signal, improve the signal-to-noise ratio (SNR). This stage contains the following actions intended for different functions:

- Background field elimination: The background field consists of the local geomagnetic field and the magnetic field produced by the ambient sources, like power grid, traffic, buildings. We are concerned with the magnetic anomaly signal from the underground objects, so the background field should firstly be removed.
- UAV interference field removal: Unlike the interference field from the external environment, UAV interference field is an inherent noise signal from system, which is related to the attitude of the drone. Herein, we propose a calibration method based on the signal correlation to separate the magnetic anomaly signal from the total field signal.
- Data gridding: It is customary to perform data gridding, which is to compute the magnetic field of regular grid nodes from the irregularly distributed sampling points by interpolation. Thus, a two-dimensional contour magnetic map is produced to reflect the abnormal distribution of the entire test area.

The aim of the magnetic data interpretation is to estimate the location of underground targets based on the spatial magnetic field data. The input data is the gridded magnetic data and a magnetic contour map (RGB image) of the test area, the YOLOv3-based Euler deconvolution method is developed to determine the 3D location of the targets.

## 3. Materials and Methods

### 3.1. Background Field Elimination

The Earth's magnetic field is a natural magnetic field, its source is classified as two categories: internal and external [32]. The magnetic field originated from the internal sources is the main component, which generally behaves as a stable dipolar field. The magnetic field originated from the external sources contains geomagnetic pulsations, magnetic bays, magnetic storms, and so on, which are generated by the complex motion of charged particles from the solar wind interacting with the Earth's ionosphere, behaving as the time-varying geomagnetic signal. The Earth's magnetic field in the local region is approximated as a uniform and stable background field for a short time. Apart from the

geomagnetic field, the observed field is affected by various high-frequency interference fields from the surrounding environment, like the 50 Hz alternating magnetic field produced by the power grid. The background field should be removed because they do not indicate any useful information about the underground targets.

The elimination of background field includes two steps, one is the low-pass filter to remove the high-frequency interference magnetic field; the other is the detrending in the spatial domain to subtract the geomagnetic field. The raw magnetic data collected along a profile and the results after eliminating the background field are displayed in Figure 3.

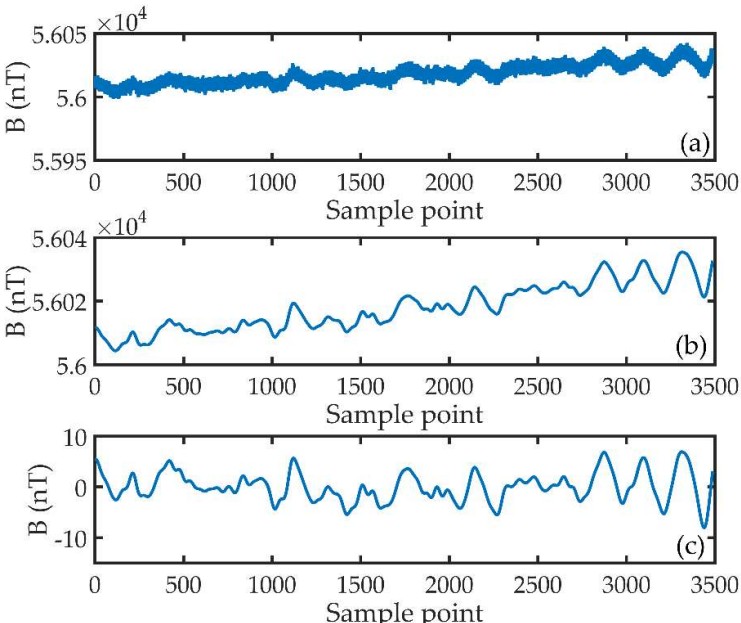

**Figure 3.** The elimination of background field. (**a**) The raw magnetic field along the profile; (**b**) The filtered magnetic field by a low-pass filter with $f_c = 3$ Hz. (**c**) The detrended magnetic field.

### 3.2. UAV Interference Field Removal

The interference field generated by the UAV platform is in the same frequency band as the anomaly field we are interested in, so it cannot be suppressed by the filter. A two-channel linear time-invariant (LTI) model is established based on the configuration of UAV-magnetometer system, as illustrated in Figure 4 [30].

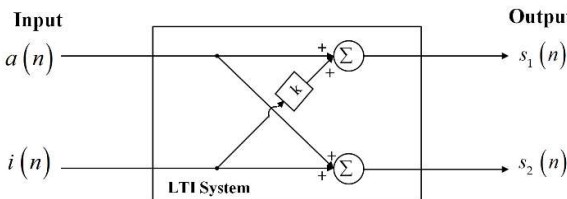

**Figure 4.** The signal model for the UAV-magnetometer system.

All signal is sampled at the sampling rate of $f_s$, $n$ indicates the sampling points, the input of LTI $a(n)$ denotes the magnetic anomaly field, $i(n)$ denotes the UAV interference signal, the outputs of LTI $s_1(n)$ and $s_2(n)$ are the measured total field signals by the magnetometer 1 and 2, respectively. The magnetometer 1 is closer to the interference sources than magnetometer 2; therefore, the interference

field measured by magnetometer 1 is $k$ times the interference field measured by magnetometer 2. The signal model is expressed as follows:

$$s_1(n) = a(n) + k * i(n) \\ s_2(n) = a(n) + i(n) \tag{1}$$

The differential signal between two sensors $d(n)$ is:

$$d(n) = (k-1) * i(n) \tag{2}$$

The interference signal is uncorrelated with the magnetic anomaly signal, so the outputs of correlating $d(n)$ with $s_1(n)$. and $s_2(n)$ are:

$$r_1 = \sum_{i=1}^{N} s_1(n)d(n) = k(k-1)\sum_{i=1}^{N} i^2(n) \\ r_2 = \sum_{i=1}^{N} s_2(n)d(n) = (k-1)\sum_{i=1}^{N} i^2(n) \tag{3}$$

Consequently, the transfer function of $k$ is estimated as:

$$k = \frac{r_1}{r_2} \tag{4}$$

Then, the magnetic anomaly field and the interference field are separated from the total signal:

$$i(n) = \frac{d(n)}{k-1} = \frac{s_1(n)-s_2(n)}{k-1} \\ a(n) = s_2(n) - i(n) = \frac{ks_2(n)-s_1(n)}{k-1} \tag{5}$$

The results after removing the UAV interference field are illustrated in Figure 5, Figure 5a,b depict two raw magnetic field signals sampled synchronously by two magnetometers (The background magnetic field is removed first), Figure 5c,d depicts the extracted interference field and magnetic anomaly field. The magnetic anomaly field and UAV interference field is superimposed and the UAV interference field from the magnetometer 1 even masks the anomaly signal completely. After the compensation, the magnetic anomaly signal and the interference field are separated from the total field, the extracted anomaly field indicates two potential targets, which is in agreement with the actual scene.

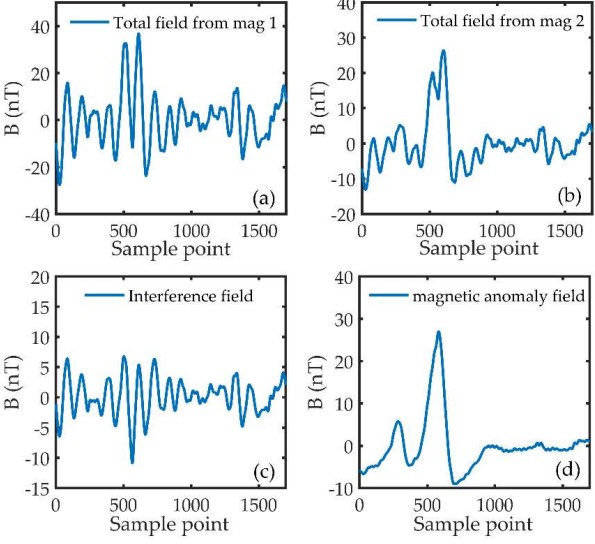

**Figure 5.** The removal of the UAV interference field. (**a**) The outputs of magnetometer 1; (**b**) The outputs of magnetometer 2; (**c**) The separated UAV interference field; (**d**) The separated magnetic anomaly field.

### 3.3. Data Gridding

Although the flight path for the drone has been planned in advance, the sampling points are still irregularly distributed on account of the restrictions of flight control accuracy and the effect of wind. Gridded magnetic data is the premise of magnetic mapping. The process of data gridding is to interpolate the magnetic field on the regular grid nodes based on the actual observation points. There are well-established data interpolation methods for the magnetic survey, such as kriging [33–35], spline [36], minimum curvature [37], and so forth. Ultimately, the gridded magnetic data over the test area and a 2D magnetic contour map (RGB image) can be obtained.

### 3.4. YOLOv3-based Euler Deconvolution

The interpretation of magnetic data is based on an improved Euler deconvolution that is the combination of YOLOv3 and Euler deconvolution. YOLOv3 is used to pick up the anomaly field automatically, Euler deconvolution is used to identify the location of buried objects based on the magnetic data chosen by YOLOv3. The complete routine of YOLOv3-based Euler deconvolution and the comparison with the traditional sliding window (SW)-based Euler deconvolution are presented below.

#### 3.4.1. Euler Deconvolution

The magnetic anomaly field satisfies the Euler's homogeneity equation, described in Cartesian coordinate as follows [24–27]:

$$\frac{\partial T}{\partial x}(x - x_0) + \frac{\partial T}{\partial y}(y - y_0) + \frac{\partial T}{\partial z}(z - z_0) = -NT(x - x_o, y - y_o, z - z_o), \tag{6}$$

where $(x_0, y_0, z_0)$ denotes the location of anomaly source, $(x, y, z)$ denotes the location of observation point, $T, \partial T/\partial x, \partial T/\partial y, \partial T/\partial z$ denote the anomalous field and its gradient fields in -$x$, -$y$, -$z$ direction. $N$ is the structure of index (SI), characterizing the attenuation rate of the amplitude of the anomalous field with the distance, which depends on the type of the sources. For the detection of near-surface targets, the anomaly source such as UXO is approximated to a point-like dipole, the SI $N$ is generally between 2.5 and 3 [24–27].

Assuming a prior value $N$, the location of anomaly source can be solved by establishing the linear equation from multiple observation points above the target, as given in (7), (8):

$$X = \left(A^T A\right)^{-1} A^T b, \tag{7}$$

where

$$A = \begin{bmatrix} \frac{\partial T_1}{\partial x} & \frac{\partial T_1}{\partial y} & \frac{\partial T_1}{\partial z} \\ \frac{\partial T_2}{\partial x} & \frac{\partial T_2}{\partial y} & \frac{\partial T_2}{\partial z} \\ \vdots & \vdots & \vdots \\ \frac{\partial T_n}{\partial x} & \frac{\partial T_n}{\partial y} & \frac{\partial T_n}{\partial z} \end{bmatrix}, b = \begin{bmatrix} x_1\frac{\partial T_1}{\partial x} + y_1\frac{\partial T_1}{\partial y} + z_1\frac{\partial T_1}{\partial z} + NT_1 \\ x_2\frac{\partial T_2}{\partial x} + y_2\frac{\partial T_2}{\partial y} + z_2\frac{\partial T_2}{\partial z} + NT_2 \\ \vdots \\ x_n\frac{\partial T_n}{\partial x} + y_n\frac{\partial T_n}{\partial y} + z_n\frac{\partial T_n}{\partial z} + NT_n \end{bmatrix}, X = \begin{bmatrix} x_0 \\ y_0 \\ z_0 \end{bmatrix}. \tag{8}$$

In theory, the observation points can be selected arbitrarily, as long as the anomaly field at the observation points is generated by an isolated source.

#### 3.4.2. YOLOv3

YOLOv3 is the third generation YOLO algorithm, is a more accurate, stable real-time target detection method, which is widely used in the image and video detection. Compared with the previous YOLO and YOLO9000, the major improvements of YOLOv3 are the category prediction changed from the single label to the multi-label, and the prediction using multiple scale fusion methods, which are the main reasons we choose YOLOv3, for more detailed information about the YOLOv3, we refer to [31].

Here, the input image is the magnetic contour map of the survey area. Unlike the generic object detection, YOLOv3 is used to detect the anomalous field generated from ferrous targets. The ferrous target is approximately a magnetic dipole, when the distance between sensor and target is more than three times the maximum size of target. The anomaly field produced by a magnetic dipole is the bipolar anomalous field with positive and negative anomalies typically. Considering the existence of noise or other interference fields, the measured anomalous field might have only one positive anomaly or one negative anomaly in the actual tests, rather than the bipolar anomaly field theoretically. Consequently, the targets to be tested are divided into two categories: positive anomaly (labelled as Positive), and negative anomaly (labelled as Negative).

Besides, the size of the window that circles the anomalous field needs to be carefully assessed, here are some strategies that need to be met:

- A window contains at least three observation points because there are three unknown parameters in Equation (7).
- Based on the principal of Euler deconvolution, the anomaly data within the windows must be generated from an isolated object, avoiding the effects from the adjacent sources.
- For the real data, the higher SNR of magnetic data is, the more reliable the results of Euler deconvolution are. The size of window is determined by a threshold that is defined as the ratio of the maximum and minimum values of the magnetic field in a window. The anomaly field with lower SNR is removed by the setting of threshold, which is set to 50% here.

The key points for applying YOLOv3 to the magnetic survey are as follows:

*Dataset*: All samples in the dataset are in picture format. The dataset is composed of a large number of 2D magnetic contour maps, divided into 85% training set, 15% test data set. The changing number, size, position, depth, azimuth, declination, remanence of the targets and background fields make the magnetic maps different. The training set contains 850 images with 3860 instances of labeled anomalies, the test set contains 150 images with 710 labeled anomalies. Figure 6 shows some examples of the dataset.

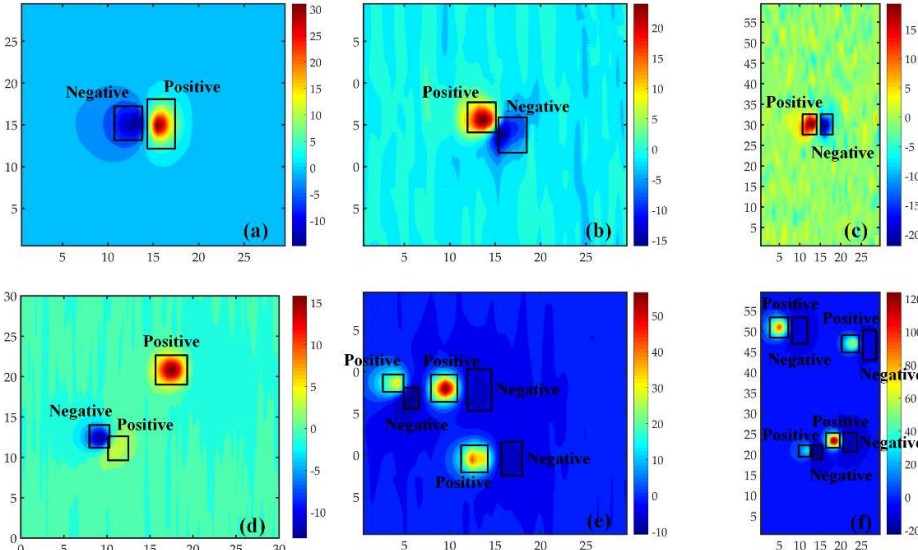

**Figure 6.** Partial samples and labelled anomalies in the dataset of You Only Look Once version 3 (YOLOv3). Each image is a 2D magnetic contour map, the unit of magnetic field in all figures is nano Tesla (nT), the targets to be detected are divided into two types: Positive and Negative. (**a**), (**b**), and (**c**) show three magnetic contour maps generated by the same target with different azimuths and declinations, remanence, and locations. (**d**), (**e**) and (**f**) show three magnetic contour maps generated by different numbers and sizes of targets, background magnetic fields.

*Detection:* YOLOv3 predicts bounding boxes at 3 different scales, the targets to be detected are classified as two classes. Batch size is set to be 12, momentum and weight decay are set to be 0.9 and 0.0005, respectively. The initial learning rate is set to be 0.0001. Finally, the output for predicted bounding box contains 6 values (top, left, bottom, right, score, class). We are more interested in the first 4 parameters, which provide the position and size of window. The 'score' reflects the confidence of the box, combining location and category confidence; the 'class' indicates the category of the predicted box.

*Evaluation:* The trained YOLOv3 yields 95.92% mAP for two classes (Positive and Negative), and the mAP is 93.87% for the test sets, indicating the reliable detection performance for the Positive and Negative anomaly.

### 3.4.3. Summary of YOLOv3-based Euler Deconvolution

The processing flow of the improved Euler deconvolution is presented in Algorithm 1. Theoretically, the Euler solutions computed by the positive anomaly field or the negative anomaly field are the same, when the anomalous field is from the same isolated target. In the actual test, if the deviation of the Euler solutions obtained by the two types of abnormal field is less than 30 cm, they can be clustered into the same target.

---

**Algorithm 1: YOLOv3-based Euler Deconvolution**

---

**Input:** Gridded magnetic data and a 2D magnetic contour map
**Output:** The location of underground targets
1: The magnetic contour map is inputted into the trained YOLOv3
2: A total of $N_{windows}$ anomaly windows are detected by YOLOv3, extract the magnetic field data inside each window
3: **for** $k = 1:N_{windows}$ **do**
4: Determine the Euler solution $X^k_{euler}$ based on the data within the window
5: **end for**
6: Cluster, if $X^i_{euler} - X^j_{euler} < 30$ cm ($X^i_{euler}$ and $X^j_{euler}$ are Euler solutions obtained by Positive and Negative anomaly windows).
7: **return** the location of potential targets, which is same as the clustered results

---

For the traditional SW-based Euler deconvolution, a window slides as a fixed step and one moving window gets one Euler solution until the test area is covered totally. All Euler solutions are clustered into the potential targets and the fake targets are filtered out by the threshold of $N$, the workflow of SW-based Euler method is summarized as Figure 7. It is worth noting that the size of window is not a fixed value in the SW-based Euler method, it changes from small to large ensuring that targets of different depth and size can be detected adaptively [24–27]. Because the sliding window is not selective, the results of Euler deconvolution depends only on the decay rate of magnetic field.

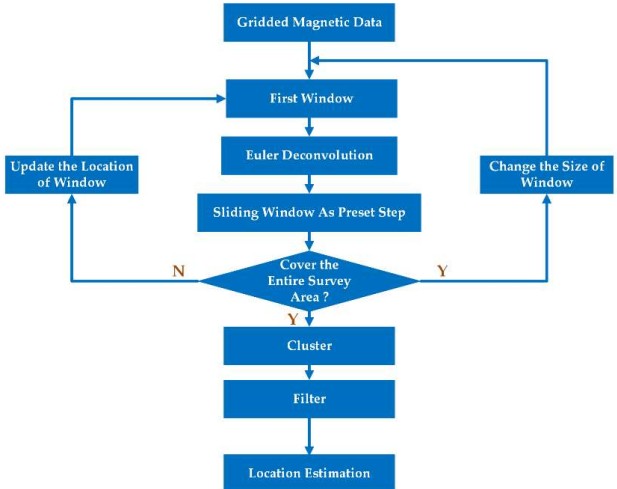

**Figure 7.** The workflow of sliding window (SW) Euler deconvolution [24], the size of windows varies from 3 to 25 grid points, and the "Filter" means filtering out the Euler solutions whose structure index *N* is less than the threshold 2.2.

By contrast, the automatic selection of windows based on YOLOv3 has considerable advantages over the sliding window. Firstly, the windows obtained by YOLOv3 are related to the strength of the magnetic field, overcoming the problem of false targets as a result of the SW method relies only on the attenuation characteristics of the magnetic field. Secondly, the size of windows chosen by YOLOv3 changes intelligently with the targets, a larger window is detected for the deeper objects, and a smaller window is detected for the shallower objects. Whereas adjusting the size of windows is clumsy for the SW Euler deconvolution, the sliding window must cover the entire area before changing the window size. The size of sliding window changes from 3 to 25 grids in the SW-Euler method, which is an empirical choice. The actual size of windows has a relation to the size of the survey area. Thirdly, the magnetic field in the windows is high SNR according to the proposed strategies, ensuring the accuracy of the positioning results. Furthermore, the Euler deconvolution is performed in the anomaly windows predicted by YOLOv3 rather than in the entire test area, simplifying the processing flow greatly, and the YOLOv3 is able to detect the anomaly targets in real-time, so the improved Euler method is faster than the SW way, which is more suitable for the large-scale magnetic survey.

## 4. Field Experiment

### 4.1. UAV-Borne Magnetic Survey

We have tested the UAV-magnetometer system and carried out the field experiments for the detection of near-surface targets in Hebei, China. The complete workflow proposed in this paper has been performed and the interpretation results were also given.

A 28 m × 33 m rectangular area was selected as the survey area, five ferrous objects with different diameters and heights were pre-buried into the test area. The programmed flight profiles run along the south-north direction, length is 62 m, spacing is 0.5 m. On a clear day, the UAV-magnetometer system was set to fly automatically as the planned profiles with AGL 2 m, the flying speed was 1.9 m/s. Figure 8 shows the flight path of the drone, the actual spacing between adjacent profiles is 0.3–0.7 m, limited by the effect of wind and the positioning accuracy of UAV.

The WGS-84 coordinate was transformed into the local cartesian coordinate system along the north, east and down, the origin was the starting point of the flight. The effective profiles above the region of interest has been obtained after cutting off the undesired flight and curved data (Figure 8).

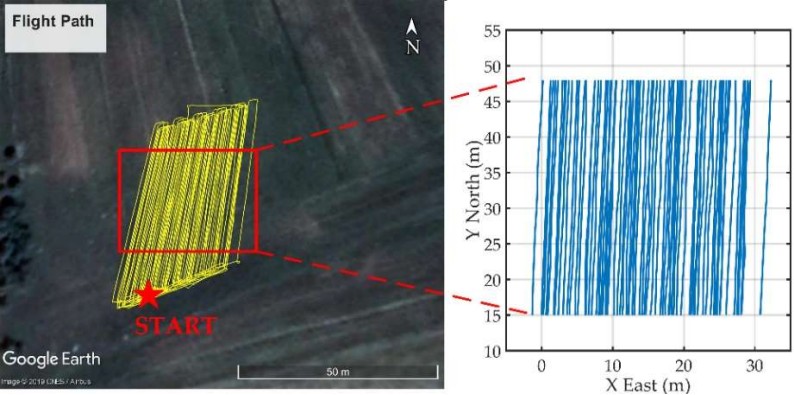

**Figure 8.** The planning of test site and flight path, multiple parallel profiles along the south-north direction cover the test area. The "START" is the starting point of the drone in the experiment and the trimmed profiles in the local coordinate are given on the right.

*4.2. Results*

The raw magnetic data was collected continuously by two magnetometers $f_s = 160$ Hz), and the 2D magnetic contour map is produced by the Kriging interpolation in Figure 9. Four features are observed from the magnetic maps of raw data: (1) The measurement inconsistency between adjacent lines results in leveling errors [38], which is related to the attitude information of the drone; (2) A subtle linear trend exists along the profile from south to north; (3) The noise signal measured by magnetometer 1 is higher than the noise measured by magnetometer 2; (4) Only two magnetic anomalies are visible, other magnetic anomalies are masked.

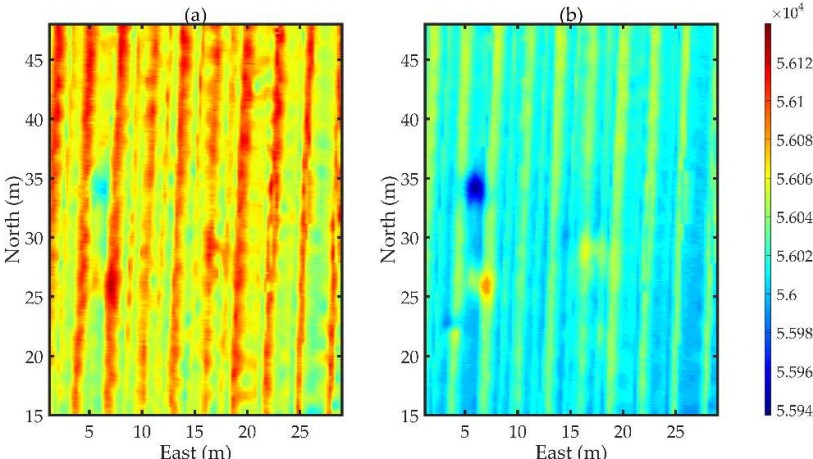

**Figure 9.** The magnetic map of raw magnetic field. (**a**) Magnetometer 1; (**b**) Magnetometer 2. (The unit of magnetic field in all figures is nano Tesla (nT)).

A lowpass filter with $f_c = 3$ Hz was used to eliminate the high-frequency interference signal firstly, then the trend term was removed to subtract the regional geomagnetic field, the processed results are displayed in Figure 10. Compared with the raw data, the curve is smoother in the magnetic map formed by the filtered magnetic field. The strips in the maps have disappeared, and some weak magnetic anomaly signals are further extracted. Nonetheless, the processed magnetic signal is still disturbed by the UAV interference field, especially for the outputs of magnetometer 1.

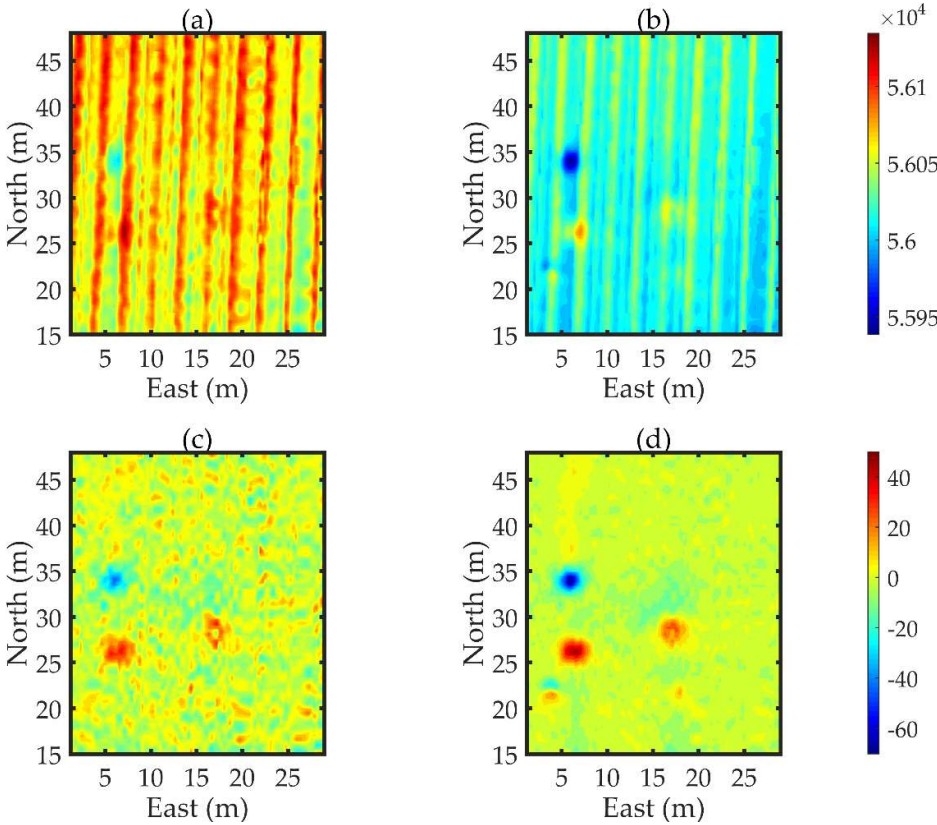

**Figure 10.** The magnetic map with the background magnetic field removed. (**a**) The filtered magnetic data from magnetometer 1; (**b**) The filtered magnetic data from magnetometer 2. (**c**) The detrended magnetic data from magnetometer 1; (**d**) The detrended magnetic data from magnetometer 2.

UAV interference magnetic field is suppressed through the proposed algorithm in Section 3.2, the compensated magnetic anomaly field is given in Figure 11. The root mean square (RMS) error is 4.9543 nT and 1.7714 nT for the total signal of magnetometer 1 and 2, respectively, the RMS of separated magnetic anomaly signal is reduced to 0.5391 nT by the proposed method, indicating that the UAV interference field has been suppressed significantly. Compared with the magnetic map of original data, the quality of magnetic map of processed signal has been improved a lot, which is able to reflect the distribution of underground anomalies clearly.

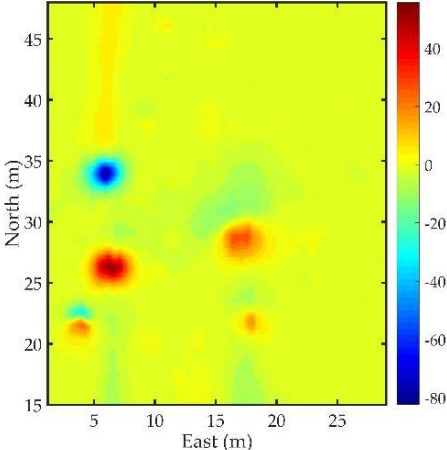

**Figure 11.** The magnetic mapping after removing the UAV interference field.

The interpretation results by the YOLOv3-based and SW-based Euler deconvolution are presented below. Firstly, the magnetic contour map (Figure 11) of the survey area is inputted into the trained YOLOv3 network, anomaly windows are detected in real-time, seven windows with the box_confidence_score greater than 0.5 are found, including five positive anomalies and two negative anomalies, as shown in Figure 12. The default SI *N* is 2.5, seven Euler solutions are determined based on the magnetic data in the seven detected windows, and finally they are clustered into five potential underground targets, as demonstrated in Figure 13a. At the same time, the same gridded magnetic data was processed by the SW-based Euler deconvolution [25], the size of sliding window changes from 3 to 25 grids, resulting in 5695 Euler solutions totally. Then, 77 Euler solutions remain after filtering out false anomalies based on the threshold of SI *N* is 2.2. Finally, 20 clustered targets are obtained with the cluster radius 50 cm, the localization results are given in Figure 13b. Five white circles '○' indicate the true position of the buried targets measured by the Real-Time Kinematic (RTK) in advance, the positioning results of the improved Euler method are consistent with the true targets. By contrast, the results of the SW Euler deconvolution contain 15 false targets, despite the filtering operation having been done.

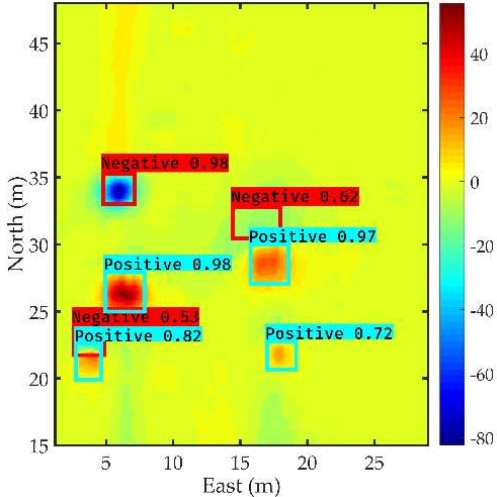

**Figure 12.** The anomaly windows detected by YOLOv3.

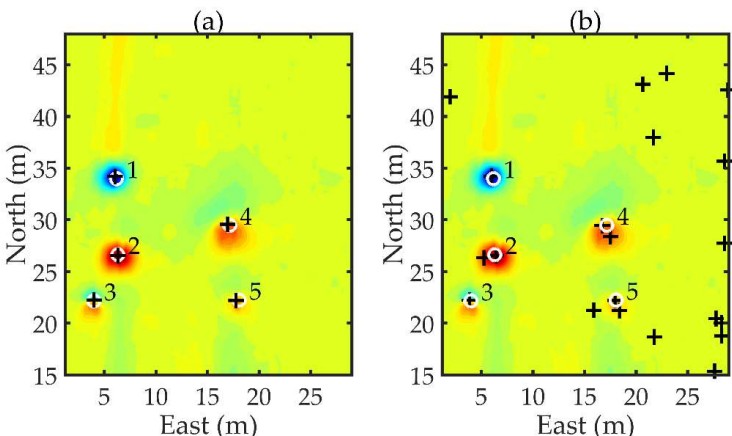

**Figure 13.** The estimated positions of two methods, white round marker '○' is the real location of buried targets measured by the Real-Time Kinematic (RTK) in advance, black cross '+' marker is the estimated location. (**a**) The YOLOv3-based Euler Deconvolution; (**b**) The SW Euler Deconvolution.

In order to evaluate the positioning accuracy of the two methods, we compare the estimated locations of five real targets in Table 1 and give the quantitative analysis of the positioning error in

Figure 14. The localization deviation of the YOLOv3-based Euler method is less than 0.3 m, which is related to the residual interference magnetic field and the truncation error caused by difference field approximation to gradient field. The positioning accuracy of the proposed method is higher than the SW-based Euler approach, especially for the depth. The depth of source is only correlated to its vertical gradient field and $N$, but the depth and $N$ are determined by the least squares at the same time in the SW method, there is mutual coupling between the two values, causing the increase of the positioning error in the depth.

**Table 1.** The comparison of real and estimated locations by two methods.

|   | True Location (m) | YOLOv3-based Euler Deconvolution (m) | SW-based Euler Deconvolution (m) |
|---|---|---|---|
| 1 | (6.17, 33.96, 0.8) | (6.03, 34.19, 0.70) | (5.98, 34.20, 0.11) |
| 2 | (6.31, 26.61, 1.0) | (6.32, 26.52, 0.75) | (6.45, 26.52, 0.21) |
| 3 | (3.99, 22.15, 0.1) | (4.01, 22.24, 0.10) | (3.85, 22.22, -0.40) |
| 4 | (17.16, 29.47, 1.1) | (16.95, 29.54, 1.06) | (16.72, 29.43, 0.11) |
| 5 | (18.07, 22.21, 0.5) | (17.78, 22.18, 0.56) | (17.96, 22.18, -0.35) |

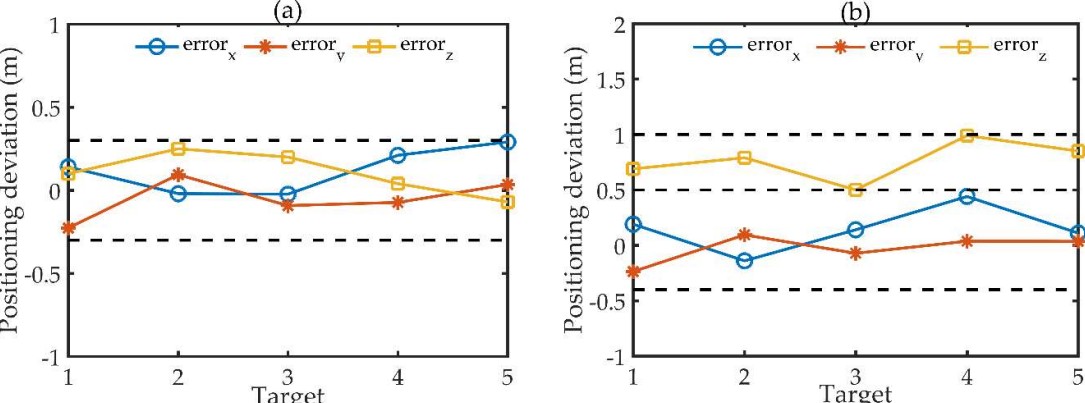

**Figure 14.** The positioning error of two methods. (**a**) The YOLOv3-based Euler Deconvolution; (**b**) The SW-based Euler Deconvolution.

## 5. Discussion

The magnetic measurement system for the UAV-borne magnetic survey is introduced, two Cs optically pumped magnetometers, GPS, radar altimeter, power and acquisition module are integrated on the multi-rotor drone. This UAV-magnetometer system is superior to the traditional magnetic measurement devices in [15,16,18,19] in two ways: (1) Two magnetic sensors are placed in the vertical direction, more magnetic field information is collected, which is beneficial for studying the characteristics of the magnetic interference field generated by the drone; (2) The magnetic sensors are fixed to the drone with a short rod, which guarantees the stability of the flight system. The construction and design of UAV-magnetometer system has been verified by multiple flight tests and it could be applied to the remote detection of near-surface targets.

### 5.1. Analysis on the Processing Flow

The magnetic data is collected by the designed UAV-magnetometer system, the background magnetic field is subtracted from the raw data first. Then, the UAV interference field in the same frequency as the magnetic anomaly field is removed by the proposed compensation method. Compared with the existed scheme that a long hard rod or soft cord (3–5 m) is attached to the drones, this paper presents a more elegant and efficient operation from the perspective of signal processing, filling the gap in the UAV magnetic compensation. The experimental results (Figure 11) show that

the RMS of interference field is reduced to less than 1 nT after the compensation and the quality of UAV magnetic data is significantly enhanced, which establishes the foundation for the subsequent interpretation of magnetic data. The residual noise is most likely to related to the geological magnetic background of the test site where the magnetic background near the surface is not uniform.

The magnetic data is interpreted by the YOLOv3-based Euler deconvolution method, the location of targets is estimated automatically, no need for an experienced data interpreter to manually extract data. The experimental results indicate that the YOLOv3-based Euler approach not only can suppress the false targets, but also has higher positioning accuracy than the SW-based Euler way. The interpretation results of SW Euler deconvolution have 15 fake targets, because the results of SW Euler method depend only on the attenuation rate of magnetic field, some small interference fields that have similar attenuation characteristics as the true anomalous field may lead to the false targets, which cannot be eliminated by the threshold of $N$. By contrast, the windows picked up by YOLOv3 is selective, which is related to the strength of the magnetic field and the Euler deconvolution depends on the decay of the magnetic field within the selected windows. The improved Euler algorithm combines the YOLOv3 and Euler, so the interpretation results depend on both the amplitude and attenuation of the magnetic anomalous field, which is able to eliminate the false anomalies. On the other hand, the size of window selected by YOLOv3 has good adaptability to different objects. The size of the window is a key parameter in the interpretation process, an appropriate window not only contains abundant and effective magnetic anomaly data from one target, but also avoids too much noise data and interference field from adjacent magnetic sources. The setting of threshold in the proposed strategies makes the size of the window adaptive to targets of different depth and size, and enables the magnetic data in the window to have higher SNR, improving the location accuracy.

Meanwhile, the workflow of improved Euler deconvolution is faster and simpler. In the field experiment, the Euler deconvolution was performed 7 times for the YOLOv3-based Euler algorithm, and the Euler deconvolution was performed 5695 times for the SW-based Euler approach. If the UAV-borne magnetic survey is carried out in a larger test area, the proposed Euler deconvolution method will become more efficient because the sliding windows should cover the entire survey region. Hence, the improved Euler processing based on the YOLOv3 is more satisfying for the UAV-borne magnetic survey. Besides, this processing routine can be also extended to the terrestrial magnetic survey and aeromagnetic survey.

### 5.2. Limits of the Research and the Future Work

The magnetic data processing workflow described in this paper is designed for the remote sensing of near-surface compact targets, such as unexploded ordnance, metallic targets, and steel pipe. These objects can be regarded as the 3D point-like magnetic dipoles; therefore, the Euler solution indicates the position of the target, which is determined by one anomaly window solely. For the interpretation of magnetic anomaly field generated from 2D or 1D geological sources, the distribution of multiple Euler solutions can reflect the trend and direction of underground anomalies, so the processing procedure herein is inappropriate. In addition, the Euler deconvolution assumes that an anomaly field within a window is generated by an isolated target. If a large target and a small target are close, the anomalous fields might be completely overlapping. The results of Euler deconvolution are unreliable in this case. Some effective methods like measuring the magnetic gradient field with higher spatial resolution to separate the magnetic anomaly field of two closed targets, should be added to the processing pipeline.

The magnetic survey is performed in a uniform and stable magnetic environment; the background field is removed by a lowpass filter and detrending. But if the targets to be detected are emplaced in a strong geological noise like magnetic ore, deposits, or soils [39], separating the magnetic field generated by the interested sources and geological magnetic background remains an ongoing challenge, because the magnetic properties of the topographic features in the test areas should be investigated in advance.

In some cases, the image filtering can be employed to extract the interested signals based on spatial wavelength characteristics of signals [40].

In this paper, the interpretation result is the location of subsurface targets, the characteristics of the target itself cannot be obtained. The future work is to identify the magnetic moment, size, and shape of the objects, achieving the classification of targets, like the discrimination between UXO and clutter.

## 6. Conclusions

This paper presents the UAV-borne magnetic survey aimed at the remote detection of near-surface targets, including the system design and magnetic field data processing and interpretation methods, and a case study is given.

The system design section describes the main components of the UAV-magnetometer system. The processing pipeline consists of three phases, the first phase ensures that the magnetic field data over the test area is acquired successfully by the UAV-magnetometer system, the second phase is to improve the quality of UAV magnetic data through a series of data processing techniques, the third phase realizes the interpretation of the UAV magnetic data and provides the position of the anomalies automatically.

One of the main contributions is a compensation method for the UAV interference field, overcoming the problem of low quality of UAV magnetic data. Moreover, a novel interpretation method of UAV magnetic data is proposed for the detection of metallic targets. Traditional Euler interpretation method is based on the sliding windows that are not selective and time-consuming; the improved approach employs the state-of-the-art YOLOv3 replacing the sliding windows, showing better performance in shortening the calculation time, reducing the false targets, and improving the positioning accuracy. The designed workflow in this paper will broaden the application of UAV magnetic sensing technique, especially for the remote detection of ferromagnetic targets.

**Author Contributions:** Y.M. proposed the methodology; Y.M., X.Z., and Y.Z. designed the validation scheme; Y.M., W.X., and Y.Z. performed the field tests; Y.M. and W.X analyzed the data; Y.M. wrote the paper. All authors have read and agreed to the published version of the manuscript.

**Funding:** The research was funded by the research project "National Natural Science Foundation of China", which is supported by the Chinese government, grant number is 41604155.

**Acknowledgments:** The authors would like to thank Zhaonan Wang for his supports in computing resources.

**Conflicts of Interest:** The authors declare no conflict of interest.

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
