# Peer review of "Automatic Detection of Near-Surface Targets for Unmanned Aerial Vehicle (UAV) Magnetic Survey"

_remotesensing, doi:10.3390/rs12030452_

Round 1
Reviewer 1 Report
The paper is very well written. The current version is improved compared to the withdrawn version. The reviewer recommendations have been addressed.
On top of that, the article is in a very interesting field that is gaining momentum in UAV technology.
Clear current limitations of UAV magnetic survey are listed and commented.
It is also a very interesting implementation of YOLO AI algorithm for a novel area.
The authors also show the advantage of YOLO versus the traditional SW Euler deconvolution and the results are very promising.
The method is very interesting, the two magnetometers spaced vertically seems to provide the needed edge in such measurements.
Overall very good read, definitely recommended for publication.
Reviewer 2 Report
The manuscript describes well the work carried out. The revised document presents more details and describes the workflow well and justifies the adopted choices. More explanations were given especially regarding the suppression of the magnetic anomaly signal from the background field, the comparison between YOLOv3 based Euler approach and Sliding Window based Euler approach.
However, validation and comparison of results with 5 targets is not entirely meaningful.
There is some specific considerations for flight mission to take into account in the case of UAV-borne magnetic survey?
Keep the same color scale for figures a and b for the figure 9. The same for figure 10.
The following reference provides additional information for the study:
Jackisch R, Madriz Y, Zimmermann R, Pirttijärvi M, Saartenoja A, Heincke BH, Salmirinne H, Kujasalo JP, Andreani L, Gloaguen R. Drone-Borne Hyperspectral and Magnetic Data Integration: Otanmäki Fe-Ti-V Deposit in Finland. Remote Sensing. 2019 Jan;11(18):2084.
Author Response
Please see the attachment

This manuscript is a resubmission of an earlier submission. The following is a list of the peer review reports and author responses from that submission.
Round 1
Reviewer 1 Report
Overall this manuscript is well written and well presented. The manuscript clearly outlines objectives. Results support the conclusions and study objectives. However, there are couple issues but major concern is: research design. Research design can be improved by testing magnetic background not for evenly distributed sites but also for more complex terrains.
Some minor issues:
Line 36: Please write full terms for MEG and MCG abbreviation
Line 393-394: Correct spelling error in figure 9 caption: nano Telsa (nT): nanotesla (nT)).
Reviewer 2 Report
The paper is very well written, definitely one of the best I had a pleasure to review.
On top of that, the article is in a very interesting field that is gaining momentum in UAV technology.
On plus - clear English language, with some minor spell check error (given below)
Clear current limitations of UAV magnetic survey are listed and commented.
It is also a very interesting implementation of YOLO AI algorithm for a novel area.
The authors also show the advantage of YOLO versus the traditional SW Euler deconvolution and the results are very promissing.
The method is very interesting, the two magnetometers spaced vertically seems to provide the needed edge in such measurements.
However, what also would be curious to see is the dataset from the higher altitude or with less spacing as the UAV flight altitude, speed and spacing of profiles can be difficult to implement in industrial applications.
The positioning of the UAV could be improved by small RTK module.
Overall very good read, definitely recommended for publication.
Some spell check:
108 which was essentially
162 involved = , which involves in some
200 which generally behaves as a stable
205 the observed field
240 magnetic field is removed first
377 starting point?
383 profiles in the local coordinate are given
394 nano Tesla
398 strips in the maps have disappeared
409 by the processing procedures?
418 are found
423 solutions remain after
Reviewer 3 Report
The paper describes a UAV system with magnetic sensors as well as the processing pipeline from data acquisition to the detection of magnetic anomalies.
The novel aspects of the paper are limited.
The magnetic sensor integration onboard UAVs has been done before. The one here is still in experimental phase (e.g., from Figure 1, landing, etc).
The processing pipeline is a standard one. The addition of deep neural network (trained for numerous real-world objects/ classes, for real-time tasks on RGB images) for the detection of two classes (negative, positive) on the demonstrated (not RGB) datasets is not fully justified. YOLOv3 is not compared with other standard detection methods. Even with a threshold on the demonstrated examples the negatives and positives can be extracted, seems that there is not a need for such a deep architecture. Therefore, the novelty is not clear and justified.
The experimental setup is not convincing. The data, validation and discussion does not provide confidence for generic, systematic, accurate results.
More comments:
Abstract
UXO -- Unexploded Ordnance (UXO)
Should indicated that YOLOv3 is an object detector
“Cs optically pumped magnetometers” should indicate (model, company, country)
The radar altimeter is not discussed.
Several grammar, spelling, and punctuation corrections
Proofreading by a native English speaker is requried
eg. B. Overview of Workflow
Figure 3 is wrong? A low pass filter doesn’t have such an effect on the raw data (Figure 3 top) from ~5.6x10^4 (nT) to ~ 0 (nT) - No information about the DC removal is provided.
“True magnetic anomaly field”. Why true? The estimated you mean most probably.
Figure 6. has no colormap values
Reviewer 4 Report
The article is well structured and complete with details.
The question are:
1- with reference to paragraph 3.4.3 Summary of improved Deconvolution: in which language was the algorithm developed: Python, Matlab, C++, IDL?
2- Chapter 4 - Field Experiment, Paragraph 4.1 The UAV Magnetic Survey: in which software the data have been processed, following the procedure shown in figure 7 on page 10?